# Effect of the Peptization Process and Thermal Treatment on the Sol-Gel Preparation of Mesoporous α-Alumina Membranes

**DOI:** 10.3390/membranes12030313

**Published:** 2022-03-10

**Authors:** Danyal Naseer, Jang-Hoon Ha, Jongman Lee, Chanhyuk Park, In-Hyuck Song

**Affiliations:** 1Ceramic Materials Division, Korea Institute of Materials Science (KIMS), 797 Changwon-daero, Changwon-si 51508, Korea; danyal@kims.re.kr (D.N.); hjhoon@kims.re.kr (J.-H.H.); jmlee@kims.re.kr (J.L.); 2Department of Advanced Materials Engineering, University of Science and Technology (UST), 217 Gajeong-ro, Daejeon 34113, Korea; 3Department of Environmental Science and Engineering, Ewha Womans University, Seoul 03760, Korea; chp@ewha.ac.kr

**Keywords:** α-Al_2_O_3_, sol-gel, ultrafiltration, peptization, phase transformation

## Abstract

Compared to traditional membrane materials, alumina membranes are particularly beneficial for industrial wastewater treatment. However, the development of mesoporous α-alumina membranes for ultrafiltration applications is still a challenge due to uncontrolled pore size. In this study, we optimized the sol-gel method for the fabrication of a high-performance mesoporous α-alumina membrane. The peptization conditions (pH and peptization time) and phase transformation of boehmite were investigated to achieve better properties of the α-alumina membrane. The surface properties of the membrane were observed to be improved by reducing the system pH to 3.5 and increasing the peptization time to 24 h. The effect of sintering temperature on the phase transformation behavior, microstructures and performance of the membranes was also elucidated. An α-alumina ultrafiltration membrane with an average thickness of 2 μm was obtained after sintering at 1100 °C. The molecular weight cut-off of the α-alumina membrane, as obtained by the filtration of aqueous PEG solution, was approximately 163 kDa (12.5 nm). This is the smallest pore size ever reported for pure α-alumina membranes.

## 1. Introduction

Membrane technology has become a key separation process for water and wastewater treatment due to its ability to utilize smaller operating systems with high efficiency [1]. Notably, porous ceramic membranes have attracted considerable attention for pressure-driven membrane processes because of several advantages, including high mechanical strength, thermal and chemical stability, pressure resistance and longer lifetimes [2,3]. Alumina is one of the most common ceramic materials for membrane applications due to its intrinsic properties of hydrothermal stability [4,5], and hydrophilic nature to mitigate fouling [6]. The most efficient fabrication routes for ceramic membranes include sol-gel, dip coating and sintering processes. The sol-gel method helps to obtain colloidal solution followed by dip coating of the substrate in the prepared solution. Afterward, the coated substrate is subjected to thermal treatment resulting in the formation of the final membrane layer [7].

The sol-gel method is the most appropriate for the preparation of alumina ultrafiltration membranes, as it allows better control over structural and morphological properties [8,9], but alumina membranes are often in the γ-phase and cannot withstand the harsh conditions of industrial processes. Thus, a higher sintering temperature was adopted for the transformation of the γ- to α-phase to achieve high thermal and chemical resistance [10]. However, transformation at higher sintering temperatures (≥1200 °C) leads to grain growth and a coarser pore structure, making the synthesis of α-membranes a challenging process.

The formation of a well-defined mesoporous α-Al_2_O_3_ membrane using the sol-gel method critically depends on the amount of water for the hydrolysis reaction, peptization (pH value), and temperature [11,12,13]. Peptization is the process responsible for the dispersion of agglomerates to form a colloidally stable suspension by mixing it with a small amount of electrolyte which is called a peptizing agent [14]. The peptizing effect is mainly dependent on the type and concentration of the peptizing agent [15,16]. When added to the sol, the peptizing agent dissociates into its ions and adsorb on the surface of the particles to form an organic layer inducing the repulsive forces among them. The concentration of peptizing agent and peptization time determine the thickness of the organic layer. As the adsorbed organic layer grows, the distance separating the particles increases resulting in the formation of large pores and vice versa [17]. In the case of boehmite sol, the optimized peptizing conditions cause the fracture of boehmite agglomerates, resulting in the formation of boehmite sol particles of smaller average diameter with a homogenous dispersion. Upon heat treatment in an air atmosphere, boehmite undergoes a topotactic transformation into transitional alumina, and the size of boehmite particles is reflected in those final alumina phases [18].

In most works on the fabrication of alumina membranes, the commonly used peptizing agents include nitric acid, acetic acid and hydrochloric acid. In comparison to acetic acid, nitric acid and hydrochloric acid have been regarded as the strongest acid that leads to the effective peptization of the boehmite sol [19]. Wi et al. [20] fabricated an α-Al_2_O_3_ membrane by using nitric acid as a peptizing agent with a mean pore size of 80–100 nm at 1250 °C. The emphasis was on the control of the sol viscosity to avoid infiltration as a function of the concentration of the peptizing agent. Schaep et al. [21] adopted a sol-gel method with nitric acid as a peptizing agent to produce an α-alumina membrane with a mean pore size of 67.8 nm at 1200 °C. α-Al_2_O_3_ membranes at a sintering temperature of 1100 °C were also prepared with a pore size of 75 nm by using glycolic acid as a peptizing agent [22]. Despite the extensive research, no pure α-Al_2_O_3_ membrane top-layer with a pore size smaller than 50 nm has been reported at a sintering temperature of ≤1100 °C. The reason is that the combined effect of the peptizing agent concentration and peptization time has not been the focus of interest in previous studies. Moreover, in recent research [23], acetic acid has been demonstrated as the most suitable peptizing agent for the stabilization of boehmite sol relative to nitric acid. It was found that acetic acid forms an organic layer on the surface of the boehmite particles that restraint their growth. Moreover, it alters the morphology of the particles in a way to provide fast permeation routes. Thus, the effect of the peptization process on the properties of alumina membranes remains unclear.

In this paper, a more systematic investigation is presented of the effect of peptization and phase transformation of alumina for the formation of mesoporous α-membranes by the sol-gel route. The influence of acetic acid as a peptizing agent on the morphological properties of the membrane was studied by varying its concentration (pH value) and peptization time. The effect of thermal treatment on the phase transformation behavior and characteristics of alumina membranes was also investigated. The properties of the membranes, including phase composition, surface morphology, membrane thickness and pore size, were well characterized. The retention of PEG and the pure water permeability of the α-alumina membrane were reported for effective functional assessment.

## 2. Materials and Methods

### 2.1. Synthesis of Boehmite Sol

The synthesis procedure of boehmite sol was as follows. Aluminum-tri-sec-butoxide (ASB, 97%, Sigma Aldrich, Darmstadt, Germany) was used as a precursor to prepare boehmite sol, together with deionized (DI) water as a hydrolysis agent and acetic acid (CH_3_CO_2_H, ≥99.7%, Sigma Aldrich, Darmstadt, Germany) as a peptizing agent. The molar ratio of aluminum-tri-sec-butoxide to DI water was maintained at 1:92.4. It is observed that aluminum alkoxide is highly reactive with water during the hydrolysis reaction. To slow down the hydrolysis, 3 mL alcohol (CH_3_CH_2_OH, ≥99.5%, Sigma Aldrich, Darmstadt, Germany) was mixed with 0.3 M Al-tri-sec-butoxide (73.8 g) via violent agitation [24]. Then, the prepared solution was added to 500 mL hot water (90 °C) and stirred at 90 °C for 1 h to hydrolyze the alkoxide. The alumina sol showed hydrolysis and polycondensation reactions for ASB [25]. For peptization, acetic acid (CH_3_COOH) was added to the mixture. Considering the isoelectric point (7.5–8.2) of boehmite as a reference, the sol was maintained at pH values of 3.5, 4.5 and 5.5 with subsequent stirring for a series of peptization times: 1, 6, 24 and 72 h. The sol-gel reaction followed the sequence given in Figure 1.

An aqueous solution of 10% PVA was then prepared by dissolving PVA powder (M.W. 31,000–50,000, Sigma Aldrich, Darmstadt, Germany) in DI water. It was used as a binder to prevent crack generation and as a viscosity regulator. For this purpose, the sol and the PVA solution were mixed at a 1:1 (*v*/*v*) ratio and stirred for 1 h to homogenize.

Multichannel α-Al_2_O_3_ porous supports of cylindrical tube-type dimensions (outer diameter: 24 mm; length: 150 mm) and 30 inner holes (inner diameter: 2 mm) were prepared in the laboratory and used as a substrate material. The average flexural strength of the alumina support layer was reported to be approximately 70 MPa [26]. Moreover, the macroporous support layer and intermediate microfiltration layer showed average pore sizes of 0.8 μm and 0.07 μm, respectively. The inner channels of the tubular α-Al2O3 substrate were dip coated using a conventional table-top dip-coater (EF-4300, E-flex, Bucheon, Korea) for 50 s with dipping and withdrawal speeds of 5 mm/s. The coated substrates were then dried at ambient temperature for 24 h. Finally, the dried membranes were calcined at 450 °C (3 °C/min) followed by sintering at various temperatures between 600 °C and 1200 °C (3 °C/min) for 2 h to investigate the phase transformation behavior.

### 2.2. Characterization

The XRD patterns were determined to identify the phases by means of an X-ray diffractometer (D/max-2200 PC, Rigaku Co. Ltd., Tokyo, Japan) at 40 kV and 200 mA, with a Cu Kα radiation source. Low voltage scanning electron microscopy (Merlin compact, Carl Zeiss, Jena, Germany) and field emission scanning electron microscopy (TESCAN MIRA, Brno-Kohoutovice, Czech Republic) were utilized to observe the microstructure of the membrane and thickness of the coatings. The particle size and micromorphology of the unsupported samples were observed by a transmission electron microscope (JEM 2100F, JEOL, Akishima, Japan). With a BET method, the specific surface area and pore size of the unsupported membranes were measured using a surface area and porosity analyzer (BELSORP-mini II, Osaka, Japan).

The retention of polyethylene glycol (PEG) experiments was carried out to determine the MWCO values by using a TOC analyzer (TOC-V CPH, Shimadzu Corp., Kyoto, Japan). The water permeability was measured using a crossflow filtration system (Membrane System, Sepra Tek, Daejeon, Korea). The operating transmembrane pressure was maintained at 1 bar, and the results were reported in units of LMH per unit bar (L m^−1^ h^−1^ bar^−1^).

## 3. Results and Discussion

### 3.1. Effect of Peptization

The main objective of this study is to obtain a high-performance α-alumina membrane with chemical and structural stability for industrial wastewater separation processes. However, cracks were observed on the membrane surface synthesized without the addition of peptizing agent and calcined at 600 °C. The pH of boehmite sol without the addition of acetic acid was observed to be 8.0, but with the increasing concentration of acetic acid, the pH tends to decrease. To obtain a defect-free membrane, it was suggested to study the effect of peptization on the surface morphology of membranes by varying the concentration of acetic acid and the peptization time. The specimens were prepared in the pH range of 3.5–5.5 under peptization time, increasing exponentially from 1 h to 72 h, as shown in Figure 2.

At pH 5.5, the membrane synthesized after a 1 h peptization time exhibited a cracked surface with particle aggregates. By continuing the peptization time, the membrane surface appears imperfect and nonuniform. It is noticeable that the membrane fabricated at pH 5.5 is dominated by defects and an inhomogeneous surface throughout the entire peptization time.

At pH 4.5, after 1 h of peptization time, the membrane appears to have a cracked surface and particle aggregates; however, after 6 h of peptization, the membrane surface overcomes the cracks but shows surface roughness. By continuing peptization for up to 72 h, a defect-free membrane is formed, and its surface appears to be much smoother.

The membrane obtained at pH 3.5 and after 1 h of peptization shows large surface cracks. After 6 h of peptization, the cracks disappear, and the surface becomes smooth without any aggregates. By continuing the peptization time up to 72 h, the membrane surface becomes more uniform and homogenous.

Comparison of the membranes prepared at different pH values demonstrates that the obtained membranes after 1 h of peptization do not appear to be very different, and all of them exhibit large cracks. These results also show that at pH 5.5, the effect of peptization time is not pronounced. However, membranes synthesized at pH 4.5 and 3.5 show a similar change from cracked and nonuniform surfaces to defect-free and smooth surfaces with continuous peptization.

In the process of peptization, acetic acid (CH_3_COOH) dissociates into cations (H^+^) and anions (CH_3_COO^−^) that adsorb on the surface of sol particles under the effect of electric charge to stabilize the boehmite sol. The boehmite materials have an isoelectric point of approximately 7.2–8.2 pH. As boehmite particles have a positive surface charge at pH values below the isoelectric point, they adsorb more anions than cations. Thus, the dissociative CH_3_COO^−^ anions of acetic acid form the first adsorption layer followed by adsorption of H^+^ cations [17,27]. In the case of a short peptization time (1 h), acetic acid does not dissociate completely into cations (H^+^) and anions (CH_3_COO^−^) due to a slow dissociation reaction [19]. This leads to the formation of an incomplete organic layer around boehmite colloidal particles. As a result, partial attractive and repulsive forces are generated among particles that destabilize the colloidal suspension. Conversely, increasing the peptization time allows acetic acid ions to form a uniform organic layer around the particles and induces strong repulsive forces among them.

Figure 3 depicts the breakdown of boehmite agglomerates into smaller particles upon the adsorption of acetic acid. However, in the case of membranes synthesized at pH 5.5, crack formation occurs regardless of the peptization time due to an insufficient concentration of acetic acid to completely adsorb on the surface of boehmite particles. Conclusively, membrane stability is highly dependent on the relationship between pH (peptizing agent concentration) and peptization time. As in our case, a membrane synthesized at pH 3.5 with a peptization time of 24 h was suggested as an optimized condition for the preparation of boehmite sol.

### 3.2. Effect of Thermal Treatment

The thermal behavior of prepared boehmite sol-gel is very crucial, so TGA results were analyzed to determine the range of sintering temperature [28]. It was observed that after 500 °C, there was no remarkable weight loss during the formation of alumina. Figure 4 shows the phase composition of the membrane sintered in the temperature range of 600–1200 °C. The thermal treatments were performed under atmospheric air for 2 h. The XRD patterns reveal that alumina has an amorphous framework from 600 °C to 800 °C with γ-Al_2_O_3_ as the main phase. At 900 °C, there existed a mixture of γ and δ-Al_2_O_3_, while at 1000 °C, θ-Al_2_O_3_ was observed with γ-Al_2_O_3_ as a mixed phase. As the sintering temperature increased to 1100 °C, the α-phase peaks became apparent. The intensities of the diffraction peaks of the (104), (113) and (116) planes detected at 2θ angles of 35.1°, 43.3°, and 57.4°, respectively, were remarkably enhanced with increasing temperature, indicating the increasing crystallinity of the α-Al_2_O_3_ phase. It is noticeable that the α-phase was detected at a lower temperature (1100 °C) than the temperature of 1200 °C, generally reported in the literature.

The phase transition of alumina particles is also associated with the growing process of crystallite size. Figure 5 shows the relationship between the calculated values of crystallite sizes at different sintering temperatures, inferred from the XRD data (Figure 4), by using the Scherrer equation [29]. It is apparent that the crystallite size increases with increasing temperature. When the temperature reaches 1100 °C, the crystallite size increases rapidly, indicating the transformation to α-alumina. Scherrer analysis is based on the assumption that crystallites have the same size and shape. However, at 1200 °C, the crystallites of the α-phase were usually irregular, as shown in Section 3.3. Thus, the crystallite size of α-Al_2_O_3_ at 1200 °C could not be determined due to the limitation of the Scherrer equation for nonuniform size distribution.

### 3.3. Microstructural Evolution

From the cross-sectional micrographs in Figure 6, it can be easily observed that the top layer was tightly integrated with the substrate under all sintering temperatures, with uniform thickness along the membrane length. The average membrane thickness of approximately 2 μm was estimated for the α-alumina membrane sintered at 1100 °C. The active membrane layer and intermediate layer can be clearly distinguished.

In Figure 7, the top surface SEM micrographs of the membranes synthesized in the temperature range of 800 °C to 1200 °C are illustrated to present the change in particle shape and surface morphology. It is apparent from the SEM images that the membranes are smooth and defect free under all sintering temperatures. At 800 °C and 1100 °C, alumina membranes with homogenous surface morphology were attained. However, sudden grain growth can be seen at 1200 °C, above the α-phase transformation temperature, leading to a heterogeneous surface morphology with lath-like particles.

Transmission electron microscopy was used to further investigate morphological changes. Figure 8 shows the TEM images of the alumina membranes sintered at 800 °C, 1100 °C and 1200 °C. A fine microstructure for γ-alumina sintered at 800 °C can be observed in Figure 8a, whereas a flaky microstructure for α-alumina sintered at 1100 °C can be observed in Figure 8c. As the temperature increases, the particle size gradually increases. The increase in particle size can be attributed to the α-phase transition process with increasing temperature. The TEM image in Figure 8b revealed a cubic crystal structure at 800 °C corresponding to γ-alumina, while a hexagonal crystal structure was detected for the membrane sintered at 1100 °C, indicating the α-phase, as shown in Figure 8d. Thus, both the XRD patterns and TEM images confirmed the successful transition of γ to the α-phase at 1100 °C. However, at 1200 °C, there is sudden grain growth, as determined from Figure 8e–f. The particles grow at the expense of smaller particles driven by the surface energy difference provided by increasing temperature under the phenomena of Ostwald ripening [30].

According to the pore evolution model [31], the particle size has a proportional relationship to the sintering temperature and average pore size. This effect of thermal treatment on pore size is provable by comparing the BET data of membranes prepared at different temperatures. The pore size increases in the range of 7–120 nm, while the BET surface area decreases from 294 m^2^ g^−1^ to 10 m^2^ g^−1^ with increasing thermal treatment from 600–1200 °C, respectively, as reported in Table 1.

The trends of surface area and pore size with increasing temperature are presented in Figure 9.

It is worth noting that the obtained average pore size for the α-Al_2_O_3_ membrane sintered at 1100 °C, 21.8 nm, is smaller than any previously reported values, as listed in Table 2. This can be attributed to the formation of boehmite particles of smaller size by the optimized conditions of peptization, a factor that was not the focus of research in previous studies.

In the literature, the major reason for the coarser pore structure of the α-Al_2_O_3_ membrane was grain growth leading to particle agglomeration at high sintering temperatures. In this work, the peptization step was controlled for the formation of boehmite sol particles of smaller size that helped to reduce the growth of particles upon γ- to α-phase transformation. The phase transformation behavior and characteristics of alumina membranes as a function of sintering temperature depict the effect of peptization on the α-Al_2_O_3_ membrane. It can be observed that with increasing sintering temperature, the particle size of the alumina membranes increases simultaneously. At 1100 °C, the α-phase formation was followed by a sharp increase in the particle size, but interestingly, there was no particle agglomeration. This result can be associated with the strong repulsive forces induced by the adsorption of the peptizing agent on boehmite particles, inhibiting interparticle contact. The conserved size of boehmite particles controls the growth and agglomeration of particles during phase transformation.

PEG-retention curves were measured for the α-Al_2_O_3_ membrane sintered at 1100 °C [32]. The membrane shows the properties of an ultrafiltration membrane with MWCO of approximately 163 kDa, as shown in Figure 10. The molecular size of PEG tracers was calculated to be 12.5 nm correlated to their molecular weight (MW in Da) by using Equation (1) [33]:d_s_ = 0.065 (MW)^0.438^(1)

Based on these retention data, the α-Al_2_O_3_ membrane can be characterized as an ultrafiltration membrane, having a pore size in the mesoporous range [34].

The pure water permeabilities of alumina ultrafiltration membranes were tested using a crossflow filtration system, as reported in Figure 11. A significant difference in the permeability of the membranes based on the sintering temperature was observed. The pure water permeability of the γ-Al_2_O_3_ membrane sintered at 600 °C (mean pore size 7.5 nm) was approximately 5.6 L m^−1^ h^−1^ bar^−1^, while the permeability of the α-alumina membrane sintered at 1100 °C with a mean pore size of 21.8 nm was 111.9 L m^−1^ h^−1^ bar^−1^. Considering that the synthesized alumina membranes have approximately similar active layer thicknesses, the difference in permeability was mainly ascribed to the promotional effect of increasing temperature on the mean pore size. The pure water permeability curve of the top active layer was stable over the period of time. This confirmed the presence of the crack-free membrane. If the top layer had been defective, the permeability would not be stable [35].

Based on these results, a mesoporous α-Al_2_O_3_ membrane sintered at 1100 °C was successfully developed on a multichannel cylindrical tube-type substrate. The supported ultrafiltration membrane had an asymmetric structure, as shown in Figure 12.

## 4. Conclusions

In the present work, mesoporous α-alumina ultrafiltration membranes were prepared by controlling the factors of the sol-gel route. It was demonstrated that peptizing conditions are effective in determining morphological properties and forming defect-free alumina membranes. Acetic acid was used as a peptizing agent that dissociates into anions and cations to form an organic layer on the surface of boehmite sol particles. First, it led to the breakdown of boehmite agglomerates into smaller particles. Second, it induced electrostatic repulsive forces among boehmite particles to inhibit interparticle contact and to induce stability. Boehmite is a topotactic precursor, so the size of boehmite sol particles was reflected in the transitional alumina. The phase transformation of alumina membranes was investigated to verify the morphological influence of boehmite particles on the final alumina phase, and the results were compared with previous studies. A pure α-Al_2_O_3_ membrane with an average membrane thickness of 2 μm and pore size of 21.8 nm was obtained at 1100 °C without defects, which is smaller than any previously reported values. It showed MWCO of approximately 163 kDA (12.5 nm) and pure water permeability of 111.9 L m^−1^ h^−1^ bar^−1^.

Finally, the optimization of parameters of the sol-gel route led to the development of an α-Al_2_O_3_ membrane with pore size in the mesoporous range. The α-Al_2_O_3_ membrane fabricated in this study exhibited a high potential for ultrafiltration applications.

## Figures and Tables

**Figure 1 membranes-12-00313-f001:**
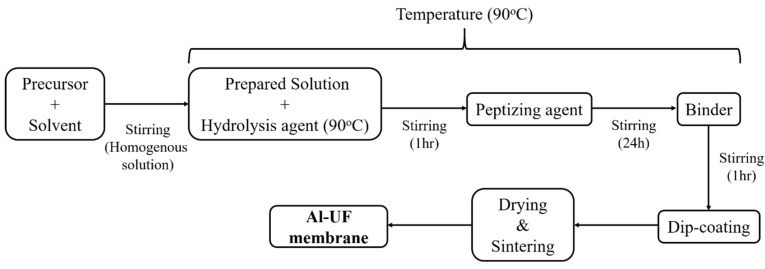
Flow diagram for the preparation of the Al_2_O_3_ ultrafiltration (UF) membrane.

**Figure 2 membranes-12-00313-f002:**
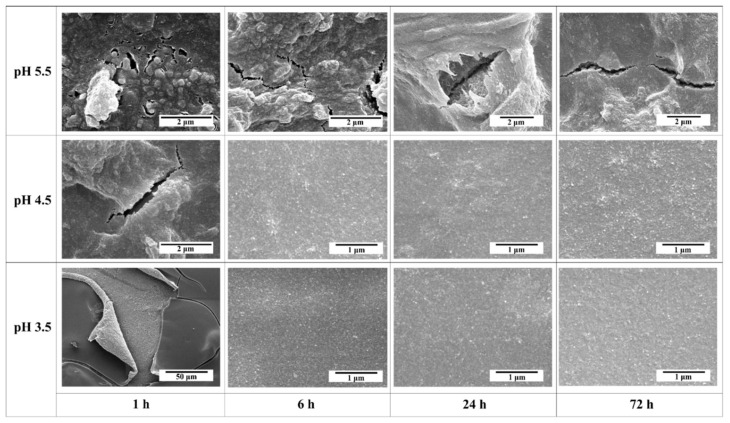
Effect of the different pH and peptization times on the appearance and stability of the Al_2_O_3_ membranes after calcination at 600 °C.

**Figure 3 membranes-12-00313-f003:**
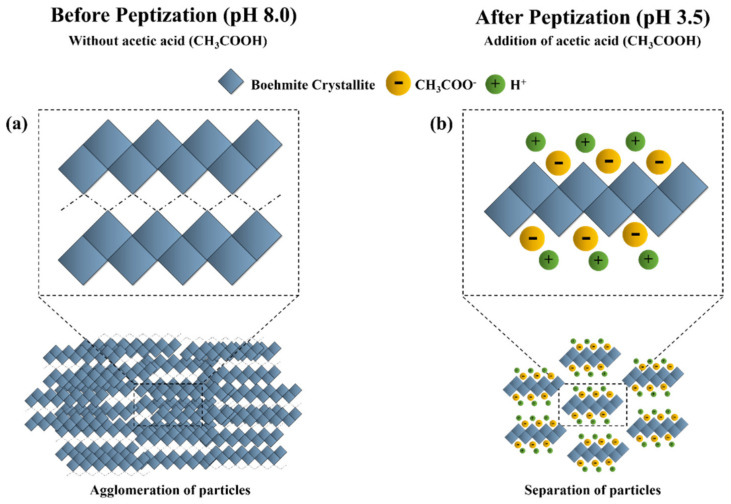
Schematic representation of the adsorption of acetic acid on boehmite crystals; (**a**) agglomeration of boehmite crystal chains due to attractive forces among them without the addition of acetic acid, (**b**) separation of boehmite crystal chains due to repulsive forces induced by the addition of acetic acid.

**Figure 4 membranes-12-00313-f004:**
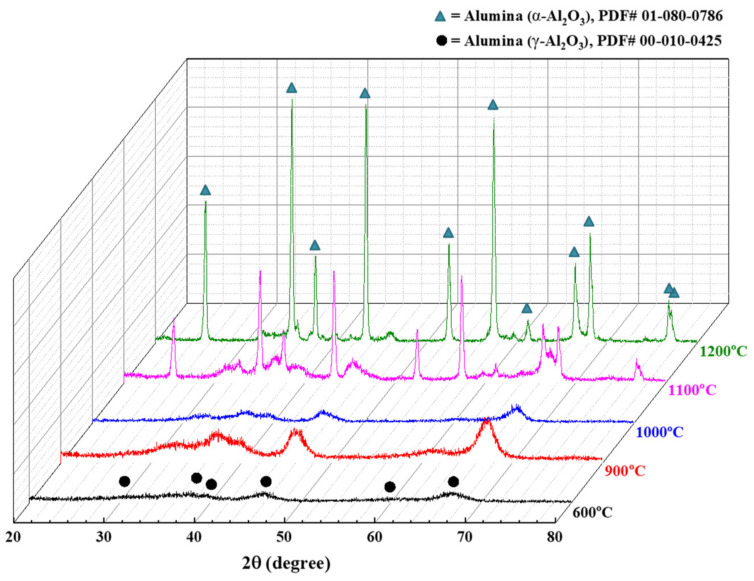
Effect of thermal treatment on the phase transformation of Al_2_O_3_ membranes (pH = 3.5 and t = 24 h).

**Figure 5 membranes-12-00313-f005:**
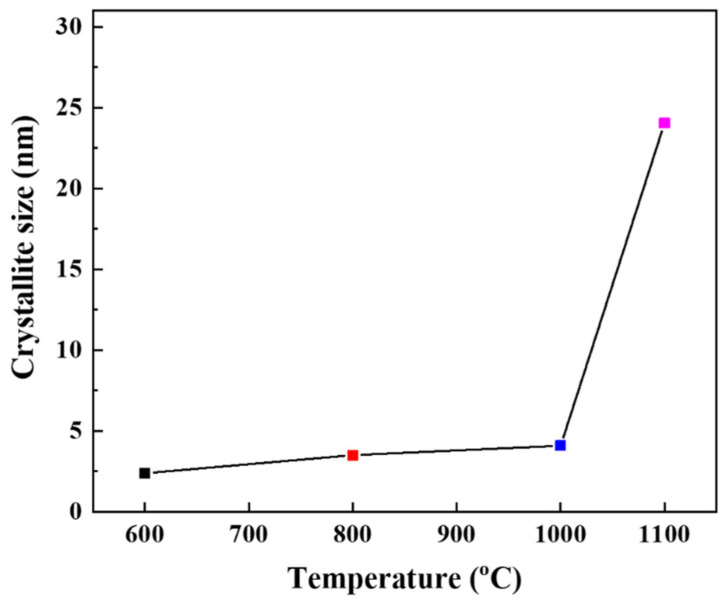
Crystallite size of alumina membranes (pH = 3.5 and t = 24 h) as a function of thermal treatment (by using the Scherrer method).

**Figure 6 membranes-12-00313-f006:**
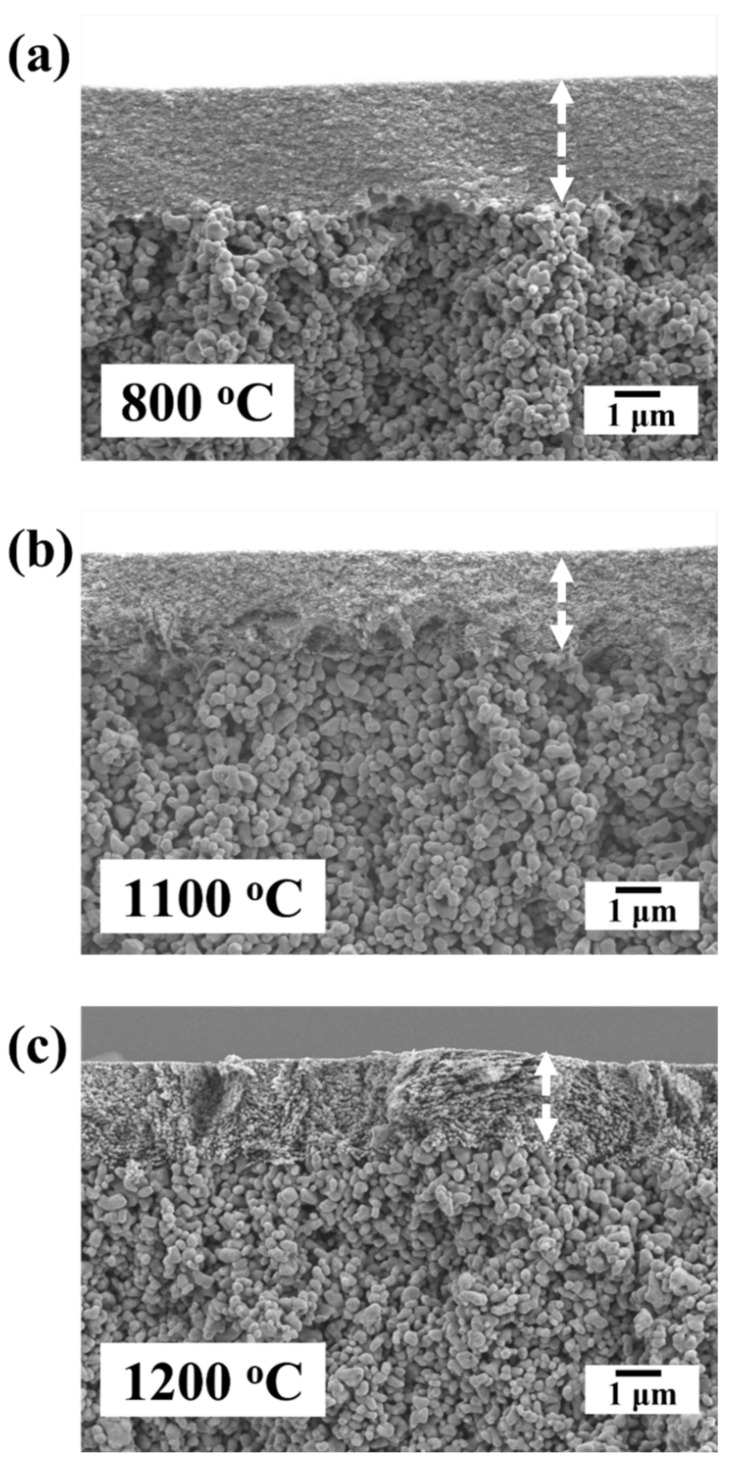
Cross-sectional SEM micrographs of the Al_2_O_3_ ultrafiltration (UF) membranes; (**a**) γ-Al_2_O_3_ membrane sintered at 800 °C, (**b**) α-Al_2_O_3_ membrane sintered at 1100 °C, (**c**) α-Al_2_O_3_ membrane sintered at 1200 °C.

**Figure 7 membranes-12-00313-f007:**
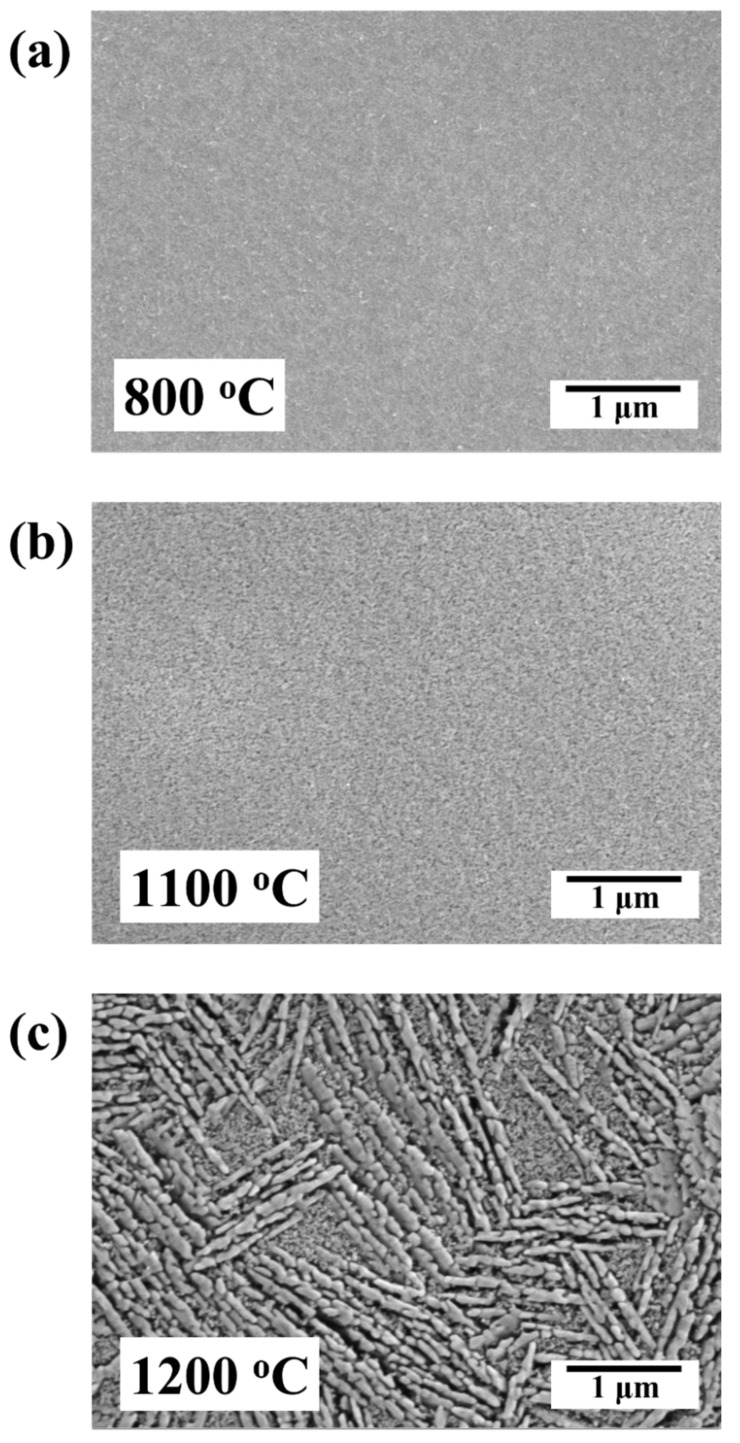
Top surface SEM micrographs of the Al_2_O_3_ ultrafiltration (UF) membranes; (**a**) γ-Al_2_O_3_ membrane sintered at 800 °C, (**b**) α-Al_2_O_3_ membrane sintered at 1100 °C, (**c**) α-Al_2_O_3_ membrane sintered at 1200 °C.

**Figure 8 membranes-12-00313-f008:**
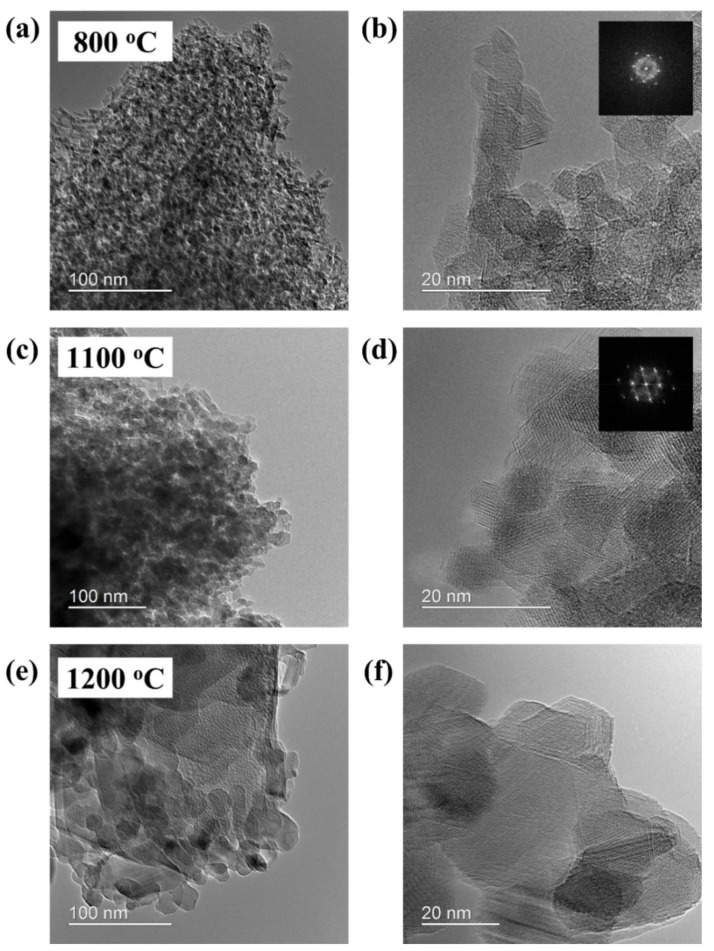
TEM images of the Al_2_O_3_ ultrafiltration (UF) membranes; (**a**,**b**) γ-Al_2_O_3_ membrane sintered at 800 °C, (**c**,**d**) α-Al_2_O_3_ membrane sintered at 1100 °C. (**e**,**f**) α-Al_2_O_3_ membrane with sudden grain growth sintered at 1200 °C.

**Figure 9 membranes-12-00313-f009:**
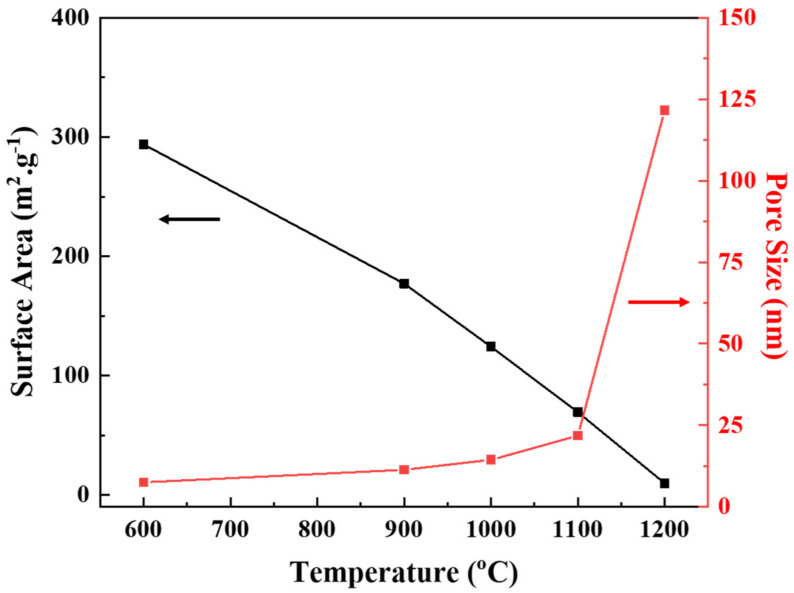
Effect of different sintering temperatures on the surface area and pore-size distributions of the Al_2_O_3_ membranes.

**Figure 10 membranes-12-00313-f010:**
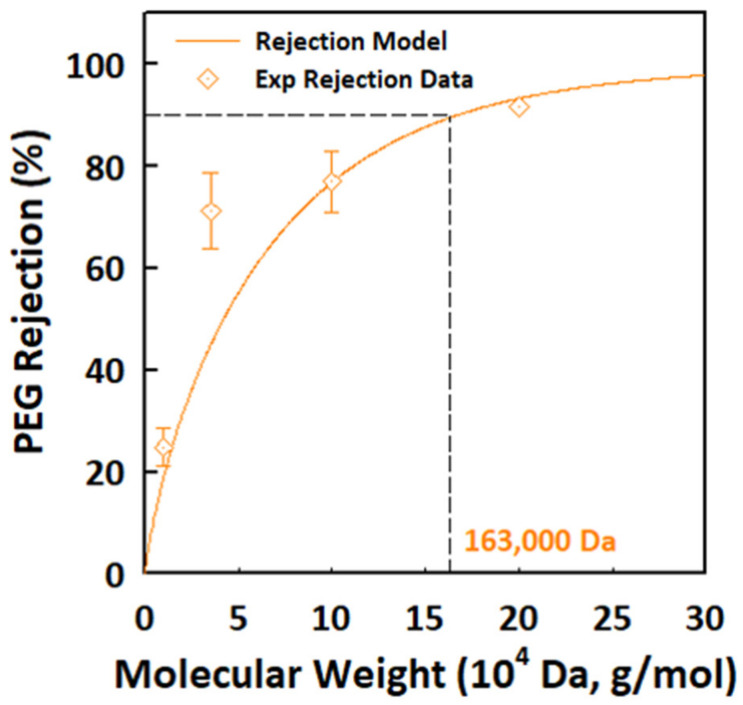
PEG retention of the α-Al_2_O_3_ (1100 °C) ultrafiltration membrane and the relationship between pore size and the molecular weight cut-off.

**Figure 11 membranes-12-00313-f011:**
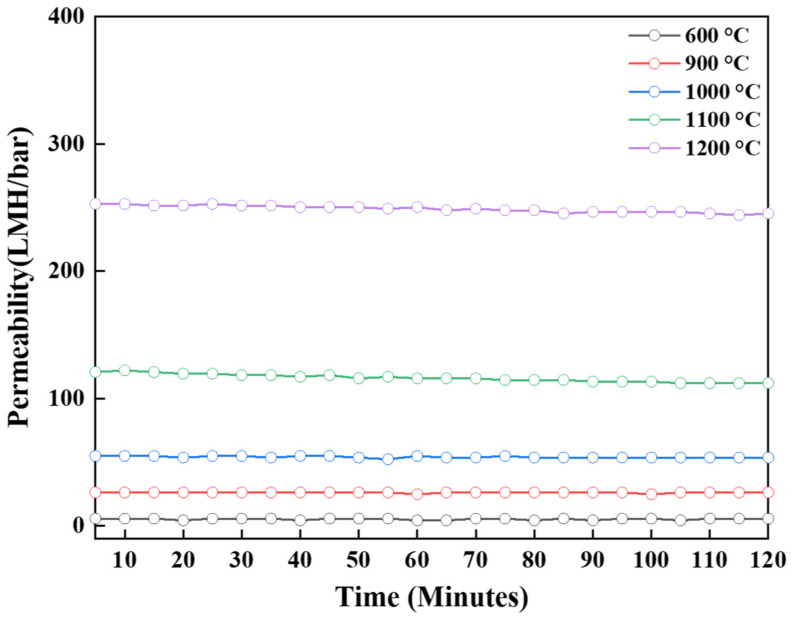
Effect of different sintering temperatures on the pure-water permeability of Al_2_O_3_ ultrafiltration (UF) membranes.

**Figure 12 membranes-12-00313-f012:**
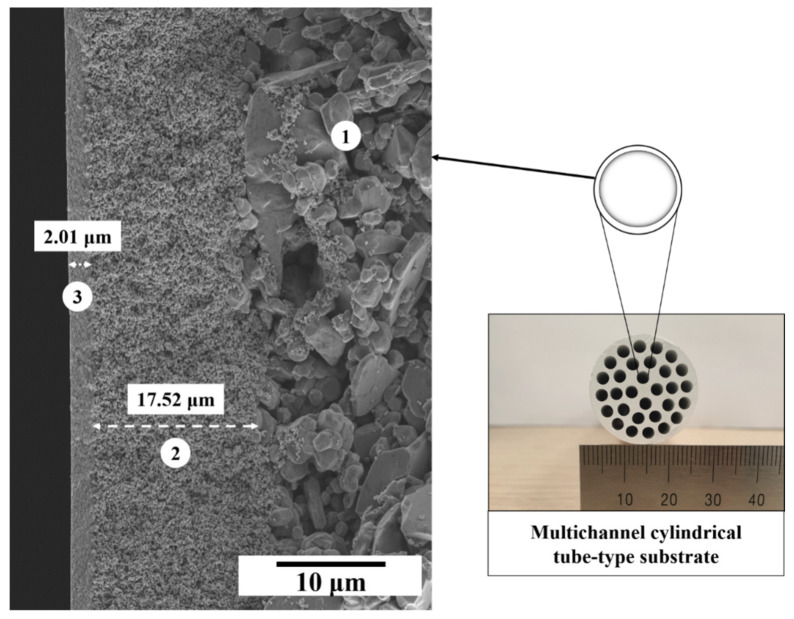
Cross-section of the asymmetric multichannel cylindrical tube-type substrate; (1) macroporous support, (2) intermediate microfiltration layer, (3) top α-Al_2_O_3_ ultrafiltration layer.

**Table 1 membranes-12-00313-t001:** Effect of the different sintering temperatures on the properties of the alumina membranes.

Temperature (°C)	Phase	Specific Surface Area (m^2^/g)	Pore Size (nm)	Permeability (LMH/bar)
600	γ	293.84	7.53	5.65
900	γ + δ	177.01	11.35	26.12
1000	γ + θ	124.17	14.43	53.55
1100	α	69.35	21.8	111.94
1200	α	9.66	121.68	245.26

**Table 2 membranes-12-00313-t002:** Textural properties of reported membranes.

Temperature (°C)	Specific Surface Area (m^2^/g)	Pore Size (nm)	Porosity (%)	Reference
1250	-	80–100	41.2	[19]
1200	6.7	67.8	29.5	[20]
1100	60	75	23	[21]
1100	69.3	21.8	37	Our work

## Data Availability

Not applicable.

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
