# Peer review of "Effect of the Peptization Process and Thermal Treatment on the Sol-Gel Preparation of Mesoporous α-Alumina Membranes"

_membranes, 2022, doi:10.3390/membranes12030313_

Round 1

Reviewer 1 Report

The article "Effect of the peptization process and thermal treatment on the sol-gel preparation of mesoporous α-alumina membranes" presents mesoporous α-alumina membrane obtained from different peptization conditions using a sol-gel method. The paper is well written; however, some improvements should be done for publication. 
1. The abstract should highlight the work´s contribution; It seems that there is no conclusion in this section. 
2. Line 60: The state of the art should be improved, and more recent works should be discussed. 
3. Line 101: The criteria for the definition of methodology should be explained. What was the base for the definition of pH and time values studied?
4. Line 173: A zeta potential analysis could corroborate the discussion?
5. The authors claim that membranes were developed for the ultrafiltration process and water or wastewater treatment. It should be interesting a discussion concerning the membrane characteristics necessary for these applications, and those obtained in the study. 

Author Response

Answer to Comments from Reviewer#1:

Respectfully, I would like to thank the reviewer to spare sometime to review this article.

  1. The abstract should highlight the work´s contribution; It seems that there is no conclusion in this section. 

Answer: I would like to thank the reviewer for suggesting the revision of abstract part. It will surely increase the importance of the manuscript. The addition has been made to the manuscript.

  1. Line 60: The state of the art should be improved, and more recent works should be discussed. 

Answer: I would like to thank to the reviewer for raising this concern. This study is specifically related to the fabrication of pure alumina membranes and to our knowledge, all the recent work related to pure alumina membranes has already been discussed in the manuscript. Some new research has been made on alumina membranes with dopants and other additives but such work was not in comparison to our study so it was not discussed.

  1. Line 101: The criteria for the definition of methodology should be explained. What was the base for the definition of pH and time values studied?

Answer: I would like to thank to the reviewer for raising this important question. The base for the definition of pH and time values were explained in detail on page 2 line 47- line 62. The main emphasis of this study is on the optimization of peptization step. Peptization is basically based on the pH and peptization time. pH can be adjusted by controlling the concentration of peptizing agent and peptization time is the time allowed for the stirring of solution after adding peptizing agent. Considering the isoelectric point of boehmite (7.5-8.2 pH) as a reference, the pH of sol suspension was experimented systematically (3.5, 4.5, 5.5 pH) and similarly the peptization time was increased exponentially (1, 6, 24, 72 h).  The addition has been made to the manuscript.

  1. Line 173: A zeta potential analysis could corroborate the discussion?

Answer: I would like to thank to the reviewer for this suggestion. Zeta potential is mainly characterized to determine the surface charge properties of the membranes. However, this study includes the optimization of sol-gel suspension to fabricate a defect-free mesoporous membrane. The isoelectric point of boehmite sol (7.5-8.2 pH) was of mere importance so just to determine the reference point to adjust the pH of suspension. However, zeta potential can be significant when dealing with the surface electrical properties of the membrane.  

  1. The authors claim that membranes were developed for the ultrafiltration process and water or wastewater treatment. It should be interesting a discussion concerning the membrane characteristics necessary for these applications, and those obtained in the study. 

Answer: I would like to thank to the reviewer for raising this issue. The target of this study is to obtain mesoporous alumina membrane with high chemical and thermal stability for industrial wastewater applications. It has been stated in previous researches that α-alumina phase is the most stable among alumina transitional phases. It has an ability to withstand vigorous environmental conditions. So, by fabricating a mesoporous α-alumina membrane, the requirements for treatment of industrial wastewater are satisfied. Moreover, based on the type of industrial wastewater such as semiconductor industry, paper industry, chemical industry, etc., the characteristics of the membrane can be further explained. However, in our case, the major focus is on the optimization of sol-gel parameters to fabricate stable α-alumina membrane, thus the characteristics based on specific applications are not discussed in detail.

Reviewer 2 Report

This manuscript reported the results on the effect of peptization and phase transformation of alumina during the preparation of alumina membrane.  It is a systematic investigation and interesting.  Here are comments.

  1. It is hard to understand that the dried membranes were calcined at 450 °C (3 °C/min) followed by sintering at various temperatures between 600 °C and 1200 °C (3 °C /min) for 2 h. What are the differences between calcination and sintering?
  2. Section 3.1. The main objective of this study is to obtain a high-performance α-alumina membrane with chemical and structural stability for industrial wastewater separation processes.  However, the appearance and stability of the Al2O3 membranes in Figure 2 do not matter with chemical and structural stability.  Please discuss.
  3. The Conclusion Section is not concise enough. There are some discusses in this conclusion.

Author Response

Answer to Comments from Reviewer#2:

We highly appreciate the reviewers’ insightful and helpful comments on our manuscript.

  1. It is hard to understand that the dried membranes were calcined at 450 °C (3 °C/min) followed by sintering at various temperatures between 600 °C and 1200 °C (3 °C /min) for 2 h. What are the differences between calcination and sintering?

Answer: I would like to thank to the reviewer for raising this concern. The key difference between calcination and sintering is that calcination is the heating of materials to remove impurities such as binder, whereas sintering is the heating of materials to bond together small particles. The sol-gel suspension optimized in this work is based on 1:1 (v/v) ratio of boehmite sol and PVA. In order to allow the complete burn-off of PVA, the membranes are initially calcined at 450 °C followed by sintering at high temperature depending on the required alumina phase. The reason for specifying this calcination temperature (450 °C) is due to the fact that decomposition temperature of PVA is 450 °C. Moreover, the dwell time of 1 h is given at this calcination temperature to allow the release of decomposed gases. If membranes are not calcined, then it can lead to the cracking of final membrane layer due to incomplete burn-off of PVA and partial release of decomposed gases.

  1. Section 3.1. The main objective of this study is to obtain a high-performance α-alumina membrane with chemical and structural stability for industrial wastewater separation processes.  However, the appearance and stability of the Al2O3 membranes in Figure 2 do not matter with chemical and structural stability.  Please discuss.

Answer: I would like to thank to the reviewer for raising this issue. The investigation conducted in section 3.1 (Figure 2) is merely related to the attainment of alumina membrane without any defects and with uniform surface by optimizing the peptization process. It has been stated in previous researches that α-alumina is the most stable alumina transitional phase. So, by successfully fabricating a defect-free α-alumina membrane, the objective of chemically and structurally stable membrane for industrial wastewater treatment is achieved.

  1.  The Conclusion Section is not concise enough. There are some discusses in this conclusion

Answer: I would like to thank to the reviewer for highlighting this issue. The main emphasis of this study is on the optimization of peptization process to attain mesoporous α-alumina membrane so it has been a focus of interest in all the sections.

The conclusion section has been further revised to clearly explain the objective of this study. The addition has been made to the manuscript.  

Reviewer 3 Report

The manuscript ‘Effect of the peptization process and thermal treatment on the sol-gel preparation of mesoporous α-alumina membranes’ is a very interesting paper and has important practical aspects. The scientific approach adopted in the study is rigorous and the paper is also well-organized. Therefore, I believe that manuscript should be published in the Membranes in present form.

Author Response

We highly appreciate the reviewers’ insightful and helpful comments on our manuscript.

Reviewer 4 Report

The paper is a rigorous research work of interest for the achievement of reducing the pore size of a-alumina membranes produced by sol-gel.
The following minor questions are posed to the authors:

- Please, could you provide a reference or describe the manufacture of the macroporous support? Both a macroporous and a microporous structure on said support are mentioned, but they are not described.
- In the last paragraph of page 3 (lines 121-133), please separate the main ideas with full stops. Many aspects of the characterization that are independent (microstructure and filtration/permeability tests) are described together in that paragraph.
- Lines 127 and 129 refer to "unsupported samples". What do you mean exactly? Please, provide more information on how unsupported samples have been shaped and sintered.
- In section 3.2, please indicate clearly in the caption of the Figures that these are the materials selected as optimal (pH=3.5 and t=24 h).
- In Figures 6.a and 6.b, improve the identification of the interface between the membrane and the support.

Author Response

Answer to Comments from Reviewer#4:

We highly appreciate the reviewers’ insightful and helpful comments on our manuscript.

  1. Please, could you provide a reference or describe the manufacture of the macroporous support? Both a macroporous and a microporous structure on said support are mentioned, but they are not described.

Answer: I would like to thank to the reviewer for highlighting this issue. The manuscript about the manufacture of macroporous support and microporous membrane has been submitted to the Journal of Advances in Applied Ceramics (Tracking ID: AAC3254). The manuscript is in the process due to which the reference was not provided.

  1. In the last paragraph of page 3 (lines 121-133), please separate the main ideas with full stops. Many aspects of the characterization that are independent (microstructure and filtration/permeability tests) are described together in that paragraph.

Answer: I would like to thank to the reviewer for raising this issue. The changes have been made to the manuscript.

  1. Lines 127 and 129 refer to "unsupported samples". What do you mean exactly? Please, provide more information on how unsupported samples have been shaped and sintered

Answer: I would like to thank to the reviewer for raising this query. Unsupported samples refer to the sol-gel suspension dried and sintered without coating on the support. After synthesis of boehmite sol, it is poured in the crucible followed by drying at ambient temperature and sintering at respective temperature. The final membrane layer is in the powder form. It is a common terminology used by the researchers concerning the fabrication of membrane. The unsupported membranes are only prepared for XRD and BET analysis.

  1. In section 3.2, please indicate clearly in the caption of the Figures that these are the materials selected as optimal (pH=3.5 and t=24 h)

Answer: I would like to thank to the reviewer for highlighting this issue. The addition has been made to the manuscript.

  1. In Figures 6.a and 6.b, improve the identification of the interface between the membrane and the support.

Answer: I would like to thank to the reviewer for raising this concern. The changes have been made to the manuscript.

Reviewer 5 Report

In my opinion, the manuscript should be revised before publication and more tests should be done. 

  1. In Part 3.1, the authors only use SEM as the characterization method. Could you please add some thermal and mechanical tests, e.g. TGA, DMA, etc?
  2. Same comment for Part 3.2.
  3. When studying PEG retention performance, could you please add data of other samples?  

Author Response

Answer to Comments from Reviewer#5:

We highly appreciate the reviewers’ insightful and helpful comments on our manuscript.

  1. In Part 3.1, the authors only use SEM as the characterization method. Could you please add some thermal and mechanical tests, e.g. TGA, DMA, etc?

Answer: I would like to thank to the reviewer for raising this concern. In section 3.1, the main emphasis is on the optimization of peptization conditions to obtain defect-free alumina membrane with uniform surface. Considering this objective, SEM was enough to verify the results. Of course, it would be better to add some thermal and mechanical tests, e.g. TGA, DMA, etc, but we are running out of time for submitting the revised paper, so please understand.

  1. Same comment for Part 3.2.

Answer: I would like to thank to the reviewer for raising this issue. The main purpose of TGA is to determine the weight loss of material during thermal treatment. However, in our case, the major concern is to investigate the transition of alumina phases despite of weight loss as characterized by XRD. Moreover, TGA analysis of alumina is readily available in previous researches that can be used a reference, if needed, as the analysis results of all the studies are similar.

  1. When studying PEG retention performance, could you please add data of other samples?  

Answer: I would like to thank to the reviewer for highlighting this issue. The main objective of this study is the fabrication of α-alumina membrane in mesoporous range, due to this reason, the PEG retention performance of only α-alumina membrane was discussed in the manuscript. However, the PEG retention performance of γ-alumina membrane was also determined as a reference but was not the part of manuscript just to avoid any confusion. The results for γ-alumina membrane sintered at 600 °C is added here, for the reviewer.

The membrane shows the properties of nanofiltration membrane with a MWCO of approximately 64,200 Da, as shown in Figure. The molecular size of PEG tracers was calculated to be 8.3 nm correlated to their molecular weight (MW in Da) by using equation (1):

ds=0.065 (MW)0.438

(1)

Round 2

Reviewer 5 Report

I appreciate the efforts made by the authors to address my questions and the manuscript looks better in the present form.  My suggestion is to add some references about the thermal and mechanical tests of α-alumina membranes. 

Author Response

# 5 Comments and Suggestion for Authors

I appreciate the efforts made by the authors to address my questions and the manuscript looks better in the present form.  My suggestion is to add some references about the thermal and mechanical tests of α-alumina membranes. 

Answer:

We highly appreciate the reviewers’ insightful and helpful comments on our manuscript.

 It will surely increase the importance of the manuscript. The addition has been made to the manuscript. Also, we have added necessary references in the revised manuscript, as commented by the reviewer.

  • Ha, J.H.; Abbas Bukhari, S.Z.; Lee, J.; Song, I.H. The membrane properties of alumina-coated alumina support layers and alumina-coated diatomite–kaolin composite support layers. Appl. Ceram.2018, 117, 1–8, doi:10.1080/17436753.2017.1369658.
  • He, Z.; Ng, T.C.A.; Lyu, Z.; Gu, Q.; Zhang, L.; Ng, H.Y.; Wang, J. Alumina double-layered ultrafiltration membranes with enhanced water flux. Colloids Surfaces A Physicochem. Eng. Asp.2020, 587, 124324, doi:10.1016/j.colsurfa.2019.124324.